# Microbiological Status of Venous Leg Ulcers and Its Predictors: A Single-Center Cross-Sectional Study

**DOI:** 10.3390/ijerph182412965

**Published:** 2021-12-08

**Authors:** Justyna Cwajda-Białasik, Paulina Mościcka, Arkadiusz Jawień, Maria Teresa Szewczyk

**Affiliations:** 1Department of Perioperative Nursing, Department of Surgical Nursing and Chronic Wound Care, Collegium Medicum in Bydgoszcz, Nicolaus Copernicus University in Torun, 85-821 Bydgoszcz, Poland; p.moscicka@cm.umk.pl (P.M.); mszewczyk@cm.umk.pl (M.T.S.); 2Outpatient Department for Chronic Wound Management, Antoni Jurasz University Hospital No. 1, 85-094 Bydgoszcz, Poland; 3Department of Vascular Surgery and Angiology, Collegium Medicum in Bydgoszcz, Nicolaus Copernicus University in Torun, 85-094 Bydgoszcz, Poland; ajawien@cm.umk.pl

**Keywords:** contamination, infection, leg ulcers, microbiology, ulceration, venous insufficiency

## Abstract

Venous leg ulcers are frequently colonized by microbes. This can be particularly devastating if the ulcer is infected with alert pathogens, i.e., highly virulent microorganisms with well-developed mechanisms of antibiotic resistance. We analyzed the microbiological status of venous leg ulcers and identified the clinicodemographic predictors of culture-positive ulcers, especially in ulcers with colonization by alert pathogens. Methods: This study included 754 patients with chronic venous leg ulcers. Material for microbiological analysis was collected by swabbing only from patients who did not receive any antibiotic treatment. Results: A total of 636 (84.3%) patients presented with culture-positive ulcers. Alert pathogens, primarily Pseudomonas aeruginosa, were detected in 28.6% of the positive cultures. In a logistic regression model, culture-positive ulcers were predicted independently by age > 65 years, current ulcer duration > 12 months, and ulceration area greater than 8.25 cm^2^. Two of these factors, duration of current ulcer > 12 months and ulceration area > 8.25 cm^2^, were also identified as the independent predictors of colonization by alert pathogens. Conclusions: Colonization/infection is particularly likely in older persons with chronic and/or large ulcers. Concomitant atherosclerosis was an independent predictor of culture-negative ulcers.

## 1. Introduction

Leg ulcers are a common chronic condition and a significant challenge for healthcare systems. The prevalence of leg ulcers (active and healed in total) is estimated to be approximately 3% [1], with 70–90% of the ulcerations being a consequence of chronic venous insufficiency [2,3,4]. Microbial colonization of venous leg ulcers is a frequent finding [5]. Factors for chronic venous insufficiency also increase the susceptibility to microbial invasion. These factors include edema, lipodermatosclerosis, hemosiderin deposition, dermatitis, atrophie blanche, and persistent proinflammatory immune responses. Venous leg ulcer patients also have propensities for limited ankle mobility, deep vein thrombosis, thrombophlebitis, impaired perfusion, and tissue necrosis. The presence of comorbidities such as obesity, arterial insufficiency, diabetes, and autoimmune diseases increases the risk of complications from colonization and microbial infection [6,7].

Research has shown that leg ulcers are colonized by both Gram-positive and Gram-negative bacteria. The most common Gram-negative microorganisms isolated in leg ulcer infections are *Pseudomonas aeruginosa* and *Escherichia coli*, while *Staphylococcus aureus* predominates among the Gram-positive microorganisms. These species also present microbiological selection to one or more antibiotics. Less frequently, samples of the following bacteria are isolated: *Proteus mirabilis*, *Klebsiella pneumoniae*, *Streptococcus agalactiae*, *Enterobacter cloacae*, *Proteus vulgaris*, *Acinetobacter baumanni*, *Morganella morganii*, *Klebsiella oxytoca*, *Citrobacter koseri*, *Citrobacter freundii*, *Coagulase-negative Staphylococcus* and *Stenotrophomonas maltophilia*. Non-healing ulcers and ulcers with a longer duration and larger wound surface have a higher microbiological diversity [8,9].

However, it should be emphasized that colonization of the ulcer does not necessarily result in clinically overt infection; in most colonization cases, the mechanisms of innate immunity can prevent microbial overgrowth, and, therefore, no symptomatic infection occurs. However, an overt infection may develop if the ulcer has been colonized by highly virulent pathogens, especially those capable of producing a biofilm; the consequence of such an infection may be delayed healing of the wound [10,11,12]. Moreover, long-term empirical antibiotic therapy may favor the selection of drug-resistant microbial strains within the wound [13,14]. This could be particularly devastating if the ulcer was infected with alert pathogens, i.e., highly virulent microorganisms with well-developed mechanisms of antibiotic resistance [15].

The aim of this single-center cross-sectional study was to analyze the microbiological status of venous leg ulcers and to identify the clinicodemographic predictors of culture-positive ulcers, especially in wounds colonized by alert pathogens.

## 2. Materials and Methods

This cross-sectional study included 754 patients with venous leg ulcers qualified for treatment at the Chronic Wound Management Unit of University Hospital No. 2 (between January 2001 and September 2012) and University Hospital No. 1 (between October 2012 and June 2019) in Bydgoszcz. The patients were eligible for the study if they presented with ankle–brachial index (ABI) values between 0.9 and 1.3 and had a diagnosis of chronic venous insufficiency confirmed on duplex scan. All patients were assessed on enrollment according to the Clinical-Etiology-Anatomy-Pathophysiology (CEAP) classification for chronic venous disease, with only C6 patients being included in the study.

The protocol of the study was approved by the local bioethics committee, and written informed consent was sought from all the participants.

### 2.1. Analyzed Clinicodemographic Parameters

The analysis included data from the medical documentation collected on enrollment. This documentation included the sociodemographic characteristics of the participants; information about comorbidities; history of chronic venous insufficiency; history of current leg ulcerations; and the location, depth, area, and number of ulcerations. Each ulceration’s depth was classified based on the degree of skin involvement, with ulcerations involving only the epidermis considered ‘superficial’ and those involving the dermis considered ‘deep’. The latter category included both ulcerations with partial involvement of the dermis and those that penetrated across the entire dermis thickness. The ulceration area (in square centimeters) was measured electronically with a Visitrak appliance. In patients with multiple ulcers, the area of the largest ulceration was considered during the analysis. The analysis also included other clinical characteristics of the ulceration: warmth, redness with a diameter greater than 2 cm, swelling, purulence/abscess, unpleasant odor, and pain. 

### 2.2. Microbiological Examination

Material for microbiological analysis was collected on enrollment only from patients who did not receive any antibiotic treatment. Before swabbing, the ulcer was cleaned of necrotic tissues, exudate, and foreign bodies, e.g., remnants of the dressing. Then, the wound was rinsed with phosphate-buffered saline (PBS). Depending on the wound’s clinical condition, the swabs were collected from the surface (superficial ulcers) or the deepest point (deep ulcers). Wound samples were collected by employing Levine’s technique. Sterile swabs were pre-wetted with sterile PBS. Then, gentle pressure was applied with the swab over an area of 1 cm^2^, applying pressure for at least five seconds (for an expressive capture of the tissue fluid). A simple swab was used, with no transport medium. Clinical swabs were placed back into the dry, sterile tube and immediately transported to the laboratory. Microorganisms from the swabs were recovered on selective media following incubation under standard conditions. The following steps in the process included [16,17]:−Shaking the tip of the swab in the phosphate-buffered saline solution;−A series of dilutions in a TSB (tryptone soya broth) medium;−Inoculation of 100 μL on the media:
Sheep Blood Agar;Sabouraud Glucose Agar;MacConkey Agar;Bile Esculin Azide Agar;
−The residual solution inoculated on CHROMagar Orientation;−Incubation for 1 day at 37 °C;−Interpretation of the results of microbial culture:
Positive if >3.7 × 10^4^ CFU/cm^2^;Positive regardless of CFU/cm^2^ if *Pseudomonas aeruginosa* or *Streptococcus pyogenes beta-haemolyticus* were detected.


### 2.3. Statistical Analysis

Statistical analysis was carried out with the Stata IC 16.1 package (StataCorp LLC, College Station, TX, USA). The normal distribution of quantitative variables was verified with the Shapiro–Wilk test. Summary characteristics of quantitative variables are presented below as descriptive statistics, i.e., arithmetic means, standard deviations, medians, lower and upper quartiles, and minimum and maximum values. Statistical characteristics of qualitative variables are shown below as numbers and percentages. The statistical significance of sociodemographic and clinical variables, as the predictors of culture-positive and alert-pathogen-positive ulcers, was verified with univariate and multivariate logistic regression analyses. The multivariate logistic regression models included the variables which turned out to be significant predictors (*p* ≤ 0.05) in univariate analysis.

## 3. Results

Demographic and clinical characteristics of the study patients are presented in Table 1. A total of 636 (84.3%) patients presented with culture-positive ulcers. Usually, the ulcer was colonized by one, two, or three microbial species. In a few cases, as many as 6–8 species of microbes were isolated.

Alert pathogens were detected in 182 patients with positive cultures (28.6%). The list of isolated alert pathogens included *Pseudomonas aeruginosa* (*n* = 126, 19.8% of positive cultures), *Acinetobacter haemolyticus* (*n* = 31, 4.9%), *Acinetobacter baumannii* (*n* = 21, 3.3%), methicillin-resistant *Staphylococcus aureus* (*n* = 20, 3.1%), beta-hemolytic streptococci groups C, F, and G (*n* = 10, 1.6%), *Candida* spp. (*n* = 8, 1.3%), *Streptococcus pyogenes* group A (*n* = 6, 0.9%), *Serratia marcescens* (*n* = 4, 0.6%), and beta-lactamase-producing *Escherichia coli* (*n* = 4, 0.6%).

Univariate logistic regression analysis demonstrated that the odds of culture-positive ulcers were significantly higher (or at a threshold of statistical significance) in patients older than 65 years, women, persons with a current ulcer lasting for more than 12 months, persons with deep or multiple ulcers, persons with swelling around the ulcer, and persons with an ulceration area > 8.25 cm^2^. Moreover, the ulcer cultures turned out to be positive in all patients with concomitant liver diseases, as well as in all participants with ulcer purulence/abscess or unpleasant odor. As these characteristics were present in all patients with culture-positive ulcers, they were not considered during further analysis. The odds of culture-positive ulcers were, in turn, significantly lower in patients with concomitant atherosclerosis or more than three comorbidities. There was no correlation between diabetes and a higher risk of wound infection (Table 2).

The variables mentioned above were analyzed together in multivariate logistic regression models. In the model including swelling, a classic sign of bacterial infection, the independent predictors of culture-positive ulcers were age > 65 years and current ulcer duration > 12 months. In another model, excluding the ‘Swelling’ variable, culture-positive ulcers were predicted independently by age > 65 years, current ulcer duration > 12 months, and ulceration area greater than 8.25 cm^2^. Regardless of the model, the odds of culture-positive ulcers were significantly lower in patients with concomitant atherosclerosis (Table 3).

Univariate logistic regression analysis identified duration of current ulcer > 12 months, purulence/abscess, unpleasant odor, swelling of the ulcer, and ulceration area greater than 8.25 cm^2^ as the significant predictors of alert pathogen isolation from the wound (Table 4).

In the multivariate analysis including all the variables mentioned above, the only independent predictors of alert pathogen isolation were purulence/abscess and unpleasant odor from the ulcer. However, in another model excluding all the classic signs of wound infection, i.e., purulence/abscess, unpleasant odor, and swelling, the isolation of alert pathogens was independently predicted by duration of current ulcer > 12 months and ulceration area > 8.25 cm^2^ (Table 5).

## 4. Discussion

In this study, the vast majority (84.3%) of the patients were found to have culture-positive ulcers. While this rate may seem high, it needs to be stressed that a positive result of a microbiological culture from an ulcer does not necessarily correspond to clinically overt infection. In the case of culture-positive ulcers, one should distinguish between asymptomatic colonization and fully symptomatic infection, with the intermediate stages of critical colonization and local infection [16,17].

Isolation of alert pathogens from the ulcer seems to be a more reliable predictor of clinically overt infection than identifying a culture-positive wound. Alert pathogens are defined as drug-resistant, life-threatening microorganisms that constitute a significant epidemiologic problem [15]. Among other microorganisms, the group of alert pathogens includes *Staphylococcus aureus*, *Pseudomonas aeruginosa*, *Enterococcus spp*., *Streptococcus pneumoniae, Acinetobacter spp., and Enterobacteriaceae*. In the present study, alert pathogens were isolated from 28.6% of culture-positive ulcers. Alert pathogens, especially Pseudomonas aeruginosa and *Staphylococcus aureus*, are commonly isolated from venous leg ulcers [4,13,14,18,19,20]. In previous studies, *Pseudomonas aeruginosa* and *Staphylococcus aureus* were isolated from 20–65% and 26–75% of chronic leg ulcers, respectively [10,13,14,21,22,23]. Thus, the isolation rates of *Pseudomonas aeruginosa* and *Staphylococcus aureus* in the present study, 19.1% and 3.1% of all positive cultures, respectively, should be considered relatively low, especially with regard to staphylococci.

Microbial colonization may impair the healing of the ulcer [24,25,26]. Interestingly, however, the relationship between microbial colonization and delayed healing was not observed in all previous studies [27,28,29,30]. This phenomenon could perhaps be explained by the fact that many microorganisms isolated from ulcers belong to saprophytic microflora of the skin [9] Usually, after colonization of the wound by such organisms, the mechanisms of innate immunity effectively prevent their overgrowth, development of clinically overt infection, and the resultant delayed healing [12]. Moreover, a few previous studies showed that, unlike other types of ulcers, venous ulcers often heal without delay, even in the case of massive bacterial colonization [31,32].

In nursing practice, the infection of the ulcer is diagnosed primarily based on its clinical manifestations [22,33]. A number of pathognomonic signs of wound infection have been described in the literature, including the presence of pain, redness, warmth, swelling, and purulence/abscess. The presence of those symptoms supports the clinical diagnosis of wound infection, and the primary objective of microbiological examination in such cases is targeted antibacterial treatment [18,34]. One study [35] demonstrated that the most common symptoms of bacterial infection of venous leg ulcers include abscess/purulence, and a sudden increase in the wound size and local temperature of the skin. However, none of the classic signs of bacterial infection mentioned above were identified as independent predictors of a culture-positive ulcer in the present study. The signs of bacterial infection, namely, purulence/abscess and unpleasant odor, were, in turn, independent predictors of ulcer colonization by alert pathogens. These findings seem to support the notion that microbial colonization of the wound does not necessarily equal its clinically overt infection.

This study also provided a few observations which seem significant in the context of wound infection prevention. Patient age > 65 years, duration of current ulcer > 12 months, and ulceration area > 8.25 cm^2^ turned out to be the independent predictors of a culture-positive wound. Two of these factors, ulcer duration > 12 months and ulceration area > 8.25 cm^2^, were also identified as independent predictors of the isolation of alert pathogens from the wound. Due to physical and/or cognitive deficits, older patients may experience self-care difficulties, which may promote microbiological contamination of the wound. In chronic ulcers, impaired perfusion and tissue necrosis are more pronounced, which may also favor microbial colonization. Finally, larger ulcers constitute a portal of entry for microorganisms, whether saprophytic or pathogenic. Thus, older persons with chronic and/or large ulcers seem to be particularly predisposed to microbial colonization/infection of the wound and, therefore, require special surveillance and education in terms of wound infection prevention.

The identification of concomitant atherosclerosis as an independent predictor of culture-negative ulcers could be considered a surprising finding. Bacterial infections are an established trigger of the atherosclerotic process [36]; thus, it cannot be excluded that, as a result of educational activities, patients with concomitant atherosclerosis presented with higher levels of health awareness in terms of wound infection prevention. Previous studies have shown that statins are protective and reduce the risk of a serious bacterial infection. Acute infections such as pneumonia, nephritis, connective tissue infections, surgical site infection, bacteremia, and sepsis were significantly less common in patients taking statins. The anti-inflammatory properties were independent of the lipid-lowering abilities of the statins. We did not evaluate the influence of statins on the course of infection, but almost all patients with hyperlipidaemia and atherosclerosis took them [37,38,39,40,41]. Statins may have resulted in a lower frequency of infections in patients with atherosclerosis. This issue should be subject to further research.

## 5. Conclusions

The majority of venous leg ulcers are culture positive; however, only a small proportion of the wounds are colonized by alert pathogens, a potential cause of symptomatic infection. Colonization/infection is particularly likely in older persons with chronic and/or large ulcers. Older persons with chronic and/or large ulcers seem to be particularly predisposed to microbial colonization/infection of the wound and, therefore, require special surveillance and education in terms of wound infection prevention.

### Limitation and Strength

An unquestioned strength of this study is its large sample size. However, the fact that the results were not adjusted for previous antibacterial treatment constitutes a substantial limitation. While the swabs were not collected from patients who received antibiotics at the time of enrollment, it cannot be excluded that the study group included some persons in whom the negative result of microbiological culture might be associated with previous antibacterial therapy, as well as those in whom empirical treatment contributed to the selection of drug-resistant strains.

## Figures and Tables

**Table 1 ijerph-18-12965-t001:** Clinicodemographic characteristics of the study patients.

Parameter	Value
Women	*n* = 485 (64.3%)
Mean (±SD) age (years)	65.7 ± 12.09
Age > 65 years	*n* = 415 (55.0%)
Median (range) duration of underlying disease (years)	24 (0–70)
Underlying disease > 20 years	*n* = 392 (52.0%)
Median (range) duration of current ulcer (months)	12 (1–504)
Duration of current ulcer > 12 months	*n* = 296 (39.3%)
Comorbidities	*n* = 647 (85.8%)
Rheumatoid arthritis	*n* = 118 (15.6%)
Arthritis	*n* = 234 (31.0%)
Diabetes mellitus	*n* = 160 (21.2%)
Atherosclerosis ^1^	*n* = 96 (12.7%)
Cardiovascular disease ^2^	*n* = 184 (24.4%)
Overweight/obesity (BMI ≥ 25 kg/m^2^)	*n* = 633 (84.0%)
Obesity (BMI ≥ 30 kg/m^2^)	*n* = 354 (46.9%)
Medial ulceration	*n* = 469 (62.2%)
Posterior ulceration	*n* = 121 (16.0%)
Anterior ulceration	*n* = 129 (17.1%)
Lateral ulceration	*n* = 216 (28.6%)
Circumferential ulceration	*n* = 24 (3.2%)
Posterior/circumferential ulceration	*n* = 141 (18.7%)
Ulcer locations ≥ 3	*n* = 58 (7.7%)
Multiple ulcerations	*n* = 357 (47.3%)
Deep ulceration	*n* = 659 (87.4%)
Median (range) ulceration area at the baseline (cm^2^)	8.25 (0.12–538)
Baseline ulceration area > 8.25 cm^2^	*n* = 373 (49.5%)
Purulence/abscess	*n* = 40 (5.3%)
Unpleasant odor	*n* = 103 (13.7%)
Redness	*n* = 472 (62.6%)
Swelling	*n* = 124 (16.4%)
Warmth	*n* = 369 (48.9%)
Pain	*n* = 646 (85.7%)

Notes: ^1^ Based on ABI values, patients with lower limb atherosclerosis were excluded from the study. ^2^ Patients with cardiac manifestations (according to NYHA or a history of myocardial infarction) and/or cerebral manifestations (a history of stroke/TIA). Abbreviations: BMI—body mass index; SD—standard deviation.

**Table 2 ijerph-18-12965-t002:** Odds ratios of culture-positive ulcers according to the clinicodemographic characteristics of the study patients; the results of univariate logistic regression analysis.

Variable	Ulcer Culture		OR	(−) 95% CI	(+) 95% CI	*p*
	Positive	Negative				
Age > 65 years	364 (87.7%)	272 (80.2%)	1.76	1.18	2.61	0.005
Female sex	419 (86.4%)	217 (80.7%)	1.52	1.02	2.27	0.039
Underlying disease > 20 years	330 (84.2%)	306 (84.5%)	0.97	0.66	1.44	0.896
Current ulcer > 12 months	270 (91.2%)	366 (79.9%)	2.61	1.64	4.15	<0.001
Rheumatoid arthritis	105 (89.0%)	531 (83.5%)	1.60	0.87	2.95	0.134
Arthritis	193 (82.8%)	443 (85.0%)	0.85	0.56	1.29	0.443
Diabetes mellitus	135 (84.4%)	501 (84.3%)	1.00	0.62	1.62	0.992
Atherosclerosis	72 (75.0%)	564 (85.7%)	0.50	0.30	0.83	0.008
Allergy	11 (73.3%)	625 (84.6%)	0.59	0.20	1.73	0.336
Cardiovascular disease	152 (82.6%)	484 (84.9%)	0.84	0.54	1.32	0.455
Kidney disease	6 (66.7%)	630 (84.6%)	0.37	0.09	1.48	0.158
Liver disease	4 (100.0%)	632 (84.3%)	not applicable			
Comorbidities	546 (84.4%)	90 (84.1%)	1.02	0.58	1.79	0.942
Multiple comorbidities	127 (78.9%)	509 (85.8%)	0.62	0.40	0.96	0.032
Overweight/obesity	538 (85.0%)	98 (81.0%)	1.33	0.80	2.20	0.268
Obesity	293 (82.8%)	343 (85.7%)	0.80	0.54	1.18	0.261
Medial ulceration	382 (83.4%)	246 (86.0%)	0.82	0.54	1.24	0.341
Posterior ulceration	105 (87.5%)	523 (83.8%)	1.35	0.76	2.42	0.31
Anterior ulceration	112 (87.5%)	516 (83.8%)	1.36	0.77	2.39	0.291
Lateral ulceration	188 (87.0%)	440 (83.3%)	1.34	0.85	2.12	0.207
Circumferential ulceration	22 (91.7%)	606 (84.2%)	2.07	0.48	8.92	0.329
Deep ulceration	564 (85.6%)	64 (75.3%)	1.95	1.14	3.34	0.015
Multiple ulcers	309 (86.8%)	325 (82.1%)	1.44	0.96	2.14	0.076
Purulence/abscess	40 (100.0%)	596 (83.5%)	not applicable			
Unpleasant odor	103 (100.0%)	533 (81.9%)	not applicable			
Redness	404 (85.6%)	230 (82.1%)	1.29	0.87	1.93	0.209
Swelling	115 (92.7%)	521 (82.7%)	2.67	1.32	5.43	0.007
Warmth	313 (84.8%)	321 (83.8%)	1.08	0.73	1.60	0.703
Pain	545 (84.5%)	90 (83.3%)	1.10	0.64	1.90	0.722
Ulceration area > 8.25 cm^2^	331 (88.7%)	305 (80.0%)	1.96	1.31	2.95	0.001

Abbreviations: OR—odds ratio; 95% CI—95% confidence interval.

**Table 3 ijerph-18-12965-t003:** Odds ratios of culture-positive ulcers according to clinicodemographic characteristics of the study patients; the results of multivariate logistic regression analysis.

Variable	OR	(−) 95% CI	(+) 95% CI	*p*
With the signs of infection				
Age > 65 years	1.78	1.17	2.70	0.007
Female sex	1.46	0.96	2.23	0.076
Current ulcer > 12 months	2.27	1.40	3.67	0.001
Atherosclerosis	0.46	0.24	0.88	0.019
Multiple comorbidities	0.92	0.52	1.60	0.759
Deep ulceration	1.61	0.92	2.84	0.097
Multiple ulcers	1.07	0.70	1.65	0.747
Swelling	2.02	0.93	4.36	0.075
Ulceration area > 8.25 cm^2^	1.39	0.88	2.21	0.158
Without the signs of infection				
Age > 65 years	1.75	1.15	2.65	0.009
Female sex	1.46	0.96	2.22	0.080
Current ulcer > 12 months	2.35	1.45	3.79	<0.001
Atherosclerosis	0.49	0.26	0.93	0.028
Multiple comorbidities	0.92	0.53	1.60	0.761
Deep ulceration	1.66	0.94	2.92	0.078
Multiple ulcers	1.09	0.71	1.69	0.683
Ulceration area > 8.25 cm^2^	1.61	1.04	2.51	0.034

Abbreviations: OR—odds ratio; 95% CI—95% confidence interval.

**Table 4 ijerph-18-12965-t004:** Odds ratios of the isolation of alert pathogens from the ulcers according to clinicodemographic characteristics of the study patients; the results of univariate logistic regression analysis.

Variable	Alert Pathogens		OR	(−) 95% CI	(+) 95% CI	*p*
	Yes	No				
Age > 65 years	104 (25.1%)	78 (23.0%)	1.12	0.80	1.57	0.501
Female sex	111 (22.9%)	71 (26.4%)	0.83	0.59	1.17	0.288
Underlying disease > 20 years	99 (25.3%)	83 (22.9%)	1.14	0.82	1.59	0.444
Current ulcer > 12 months	100 (33.9%)	82 (17.9%)	2.35	1.67	3.30	<0.001
Rheumatoid arthritis	35 (29.7%)	147 (23.1%)	1.40	0.91	2.16	0.130
Arthritis	62 (26.6%)	120 (23.1%)	1.21	0.85	1.72	0.296
Diabetes mellitus	35 (21.9%)	147. (24.8%)	0.85	0.56	1.29	0.445
Atherosclerosis	18 (18.7%)	164 (25.0%)	0.69	0.40	1.19	0.186
Allergy	6 (40.0%)	176 (23.8%)	2.13	0.75	6.06	0.157
Cardiovascular disease	40 (21.7%)	142 (25.0%)	0.84	0.56	1.24	0.376
Kidney disease	4 (44.4%)	178 (23.9%)	2.54	0.68	9.58	0.167
Liver disease	2 (50.0%)	180 (24.0%)	3.16	0.44	22.60	0.251
Comorbidities	156 (24.1%)	26 (24.3%)	0.99	0.62	1.60	0.973
Multiple comorbidities	40 (24.8%)	142 (24.0%)	1.05	0.70	1.57	0.822
Overweight/obesity	156 (24.6%)	26 (21.7%)	1.18	0.74	1.89	0.485
Obesity	94 (26.5%)	88 (22.1%)	1.28	0.91	1.78	0.151
Medial ulceration	117 (25.6%)	63 (22.0%)	1.22	0.86	1.73	0.269
Posterior ulceration	29 (24.2%)	151 (24.2%)	1.00	0.63	1.57	0.987
Anterior ulceration	34 (26.6%)	146 (23.7%)	1.16	0.75	1.79	0.498
Lateral ulceration	48 (22.2%)	132 (25.0%)	0.85	0.59	1.25	0.415
Circumferential ulceration	9 (37.5%)	171 (23.8%)	1.92	0.83	4.47	0.129
Deep ulceration	165 (25.0%)	15 (17.9%)	1.54	0.86	2.76	0.151
Multiple ulcers	91 (25.6%)	91 (23.0%)	1.15	0.82	1.60	0.420
Purulence/abscess	34 (85.0%)	148 (20.8%)	21.63	8.91	52.50	<0.001
Unpleasant odor	96 (93.2%)	86 (13.2%)	89.94	40.41	200.20	<0.001
Redness	113 (23.9%)	69 (24.7%)	0.96	0.68	1.35	0.807
Swelling	49 (39.5%)	133 (21.1%)	2.44	1.62	3.66	<0.001
Warmth	81 (21.9%)	101 (26.4%)	0.78	0.56	1.09	0.152
Pain	162 (25.2%)	20 (18.5%)	1.44	0.87	2.40	0.159
Ulceration area > 8.25 cm^2^	121 (32.4%)	61 (16.0%)	2.51	1.77	3.56	<0.001

Abbreviations: OR—odds ratio; 95% CI—95% confidence interval.

**Table 5 ijerph-18-12965-t005:** Odds ratios of the isolation of alert pathogens from the ulcers according to clinicodemographic characteristics of the study patients; the results of multivariate logistic regression analysis.

Variable	OR	(−) 95% CI	(+) 95% CI	*p*
With the signs of infection				
Current ulcer > 12 months	1.50	0.93	2.42	0.095
Purulence/abscess	22.82	8.53	61.05	<0.001
Unpleasant odor	84.55	37.06	192.86	<0.001
Swelling	0.82	0.41	1.63	0.569
Ulceration area > 8.25 cm^2^	1.22	0.74	2.01	0.441
Without the signs of infection				
Current ulcer > 12 months	2.02	1.43	2.87	<0.001
Ulceration area > 8.25 cm^2^	2.19	1.53	3.14	<0.001

Abbreviations: OR—odds ratio; 95% CI—95% confidence interval.

## Data Availability

The data presented in this study are available on request from the corresponding author. The data are not publicly available due to due to restrictions privacy.

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
