# Peer review of "Microbiological Status of Venous Leg Ulcers and Its Predictors: A Single-Center Cross-Sectional Study"

_ijerph, 2021, doi:10.3390/ijerph182412965_

Round 1
Reviewer 1 Report
Dear authors:
It has been a pleasure to review your paper “Microbiological Status of Venous Leg Ulcers and Its Predictors: A Single-Center Cross-Sectional Study” and I I think it is a great job for the sample and the results obtained. You can see below the recommendation.
this paper has a highlighting areas in its methodology, large sample and correct data analysis. It clearly shows the objective of the study and the method of obtaining the results. The introduction and discussion are based on a wide and updated bibliography.
As a weak point appears the finding of atherosclerosis as surprising when this blood circulation deficit can worsen or aggravate the infection, even establish it.
I do not agree with the sentence that contains the line 24-25-26 and 207-208. Atherosclerosis is a risk factor in predictable infection and this study refutes this idea. It is not a surprising factor or variable.
Author Response
Reply to the first comment on manuscript
“Microbiological Status of Venous Leg Ulcers and Its Predictors: A Single-Center Cross-Sectional Study”
We sincerely thank you to have taken time to evaluate our manuscript. We also thank the Reviewer of the IJERPH for useful suggestions that helped us improve our manuscript. Below we present the answer to Reviewer comment. All changes in the revised manuscript were highlighted (we used “Track Changes” function) and accompanied by adequate authors ’comments.
The sentence suggested by the reviewers has been corrected by us in the abstract and discussion:
Line 24-26: Conclusions: Colonization/infection is particularly likely in older persons with chronic and/or large ulcers. The identification of concomitant atherosclerosis as an independent predictor of culture-negative ulcers could be considered a surprising finding. Concomitant atherosclerosis was an independent predictor of culture-negative ulcers.
Line 207-8: The identification of concomitant atherosclerosis as an independent predictor of culture-negative ulcers could be considered a surprising finding. Concomitant atherosclerosis was an independent predictor of culture-negative ulcers.
Thank you for your time and valuable suggestions for our manuscript.

Reviewer 2 Report
Białasik et al. is tackling an interesting topic in the presented work.
The study that took place on 754 subjects revealed that immuno-compromised individuals are more likely to be subjected to infections, an expected outcome, however concomitant atherosclerosis was observed to be and independent predictor a fact that is rather novo therefore the study has value not only due to its large number of subjects but also due to its novel finding.
The title and abstract are well written, simple and concise.
The introduction is too brief, more data regarding the microbiota found in chronic and/or large ulcers should be stated.
Data regarding diabetes and leg ulcers should also be stated in the introduction. If such data is available regarding the subjects, then it should be included in results as well.
Line 77: state the kind of swab: dry or wet, simple or with transport media? This is actually an important factor that will later influence the microbiology assay.
Line 94-98 requires reformulation into a more concise and fluent form.
State the protocols and media used in the microbiology assays, for a work focused so much on microbial activity the microbiology part in terms of materials and methods is absent, such is unacceptable.
Author Response
Reply to the second comment on manuscript
“Microbiological Status of Venous Leg Ulcers and Its Predictors: A Single-Center Cross-Sectional Study”
We sincerely thank you to have taken time to evaluate our manuscript. We also thank the Reviewer of the IJERPH for useful suggestions that helped us improve our manuscript. Below we present the answer to Reviewer comment. All changes in the revised manuscript were highlighted (we used “Track Changes” function) and accompanied by adequate authors ’comments, as follows:
Reviewer's comment 1 We agree with the Reviewer that we have not clearly described the methods used, especially the protocols and media used in microbiological testing. It should have been explained anyway, we apologise for this omission. The data suggested by the reviewer was completed in the material and methods section:
2.2. Microbiological Examination
Material for microbiological analysis was collected on enrollment solely from patients who did not receive any antibiotic treatment. Before swabbing, the ulcer was cleaned of necrotic tissues, exudate and foreign bodies, e.g. remnants of the dressing. Then, the wound was rinsed with phosphate buffered saline (PBS). Depending on the wound's clinical condition, the swabs were collected from the surface (superficial ulcers) or the deepest point (deep ulcers). Wound samples were collected by employing Levine’s technique. Sterile swabs were pre-wetted with sterile PBS. Then, gentle pressure was applied as the swab applied over an area of 1 cm2, applying pressure for at least five seconds (for an expressive capture of the tissue fluid). A simple swab was used with no transport medium (simple swab, with no transport medium). Clinical swabs were placed back into the dry, sterile tube and immediately transported to the laboratory.
Microorganisms from the swabs were recovered on selective media, following incubation under standard conditions. The following steps in the process included:
- shaking the tip of the swab in the phosphate-buffered saline solution,
- a series of dilutions in TSB (tryptone soya broth) medium,
- inoculation of 100 μl on the media:
- Sheep Blood Agar,
- Sabouraud Glucose Agar,
- MacConkey Agar,
- Bile Esculin Azide Agar,
- and the residual solution - inoculated on CHROMagar Orientation,
- Incubation for 1 day at 37°C,
- Interpretation of the results of microbial culture:
- positive if> 3.7 x10 CFU/cm2
- positive regardless of CFU/cm2 if Pseudomonas aeruginosa or Streptococcus pyogenes beta-haemolyticus.
[We have added 2 additional references here:
- Dow G. Bacterial swabs and the chronic wound: when, how, and what do they mean. Ostomy Wound Manage 2003;49(5A):8–13.
- Angel DE, Lloyd P, Carville K, Santamaria N. The clinical efficacy of two semi-quantitative wound-swabbing techniques in identifying the causative organism(s) in infected cutaneous wounds. Int Wound J 2011; 8:176–185 doi: 10.1111/j.1742-481X.2010.00765.x.]
Reviewer's comment 2: As suggested by the reviewer, we reformulated lines 98-98 into a more concise form:
Line 94-98: A total of 636 (84.3%) patients presented with culture-positive ulcers. Usually, the ulcer was colonized by one, two or three microbial species. In single cases, as many as 6-8 species of microbial were isolated.
Reviewer's comment 3: In line with the reviewer's instructions, we have supplemented the introduction with data on the microflora of leg ulcers:
Introduction: Leg ulcers are a common chronic condition and a significant challenge for healthcare systems. The prevalence of leg ulcers (active and healed in total) is estimated at approx. 3% [1], with 70-90% of the ulcerations being a consequence of chronic venous insufficiency [2-4]. Microbial colonization of venous leg ulcers is a frequent finding [5]. Chronic venous insufficiency factors increase the susceptibility to microbial invasion. These factors include edema, lipodermatosclerosis, hemosiderin deposition, dermatitis, atrophie blanche, and persistent proinflammatory immune responses. Venous leg ulcer patients also have propensities for limited ankle mobility, deep vein thrombosis, thrombophlebitis, impaired perfusion, and tissue necrosis. The presence of comorbidities such as obesity, arterial insufficiency, diabetes, and autoimmune diseases increases the risk of complications from colonization and microbial infection [6,7].
Research has shown that leg ulcers are colonized by both gram-positive and gram-negative bacteria. The most common gram-negative microorganisms isolated in leg ulcer infections were Pseudomonas aeruginosa and Escherichia coli samples, while Staphylococcus aureus predominated among the positive microorganisms. These species also presented microbiological selection to one or more antibiotics. Less frequently, samples of Proteus mirabilis, Klebsiella pneumoniae, Streptococcus agalactiae, Enterobacter cloacae, Proteus vulgaris, Acineto baumanni, Morganella morganii, Kleibisiella oxytoca, Citrobacter koseri, Citrobacter faaphylcus-Stocociscus, and Stocylacter faaphylcus-negative, and Coagylacter faaphylcus-negative and Coagylacter faaphylecociscus negative, and Stocylacter faaphylcus-negative were isolated. Non-healing ulcers and ulcers with a longer duration and larger wound surface had a higher microbiological diversity [We have added 2 additional references here:
- Garcia T.F., Borges E.L., Junho T.O.C., Spira J.A.O. Microbiological profile of leg ulcer infections: review study. Rev Bras Enferm. 2021 Jun 18;74(3):e20190763. doi: 10.1590/0034-7167-2019-0763.
- Tuttle M.S. Association Between Microbial Bioburden and Healing Outcomes in Venous Leg Ulcers: A Review of the Evidence. Adv Wound Care (New Rochelle) 2015 Jan 1;4(1):1-11. doi: 10.1089/wound.2014.0535.].
However, it should be emphasized that colonization of the ulcer does not necessarily correspond to its clinically overt infection; in most colonization cases, the mechanisms of innate immunity can prevent microbial overgrowth, and no symptomatic infection occurs. An overt infection may develop if the ulcer has been colonized by highly virulent pathogens, especially those capable of producing biofilm; the consequence of such an infection may be delayed healing of the wound [8-10]. Moreover, long-term empirical antibiotic therapy may favor selecting drug-resistant microbial strains within the wound [11,12]. This could be particularly devastating if the ulcer was infected with alert pathogens, i.e. highly virulent microorganisms with well-developed mechanisms of antibiotic resistance [13].
The aim of this single-center cross-sectional study was to analyze the microbiological status of venous leg ulcers and to identify the clinicodemographic predictors of culture-positive ulcers, especially the wounds colonized by alert pathogens.
Reviewer's comment 3: Thank you for your suggestion on diabetes and its impact on the risk of wound infection.
We only evaluated patients with venous leg ulcers in the study. Diabetes (21%) and atherosclerosis (12.7%) were present in some of the patients, but we excluded them if they had any peripheral disorders, including PAD, angiopathy, and diabetic neuropathy. Nevertheless, these diseases can affect both the healing process and the risk of infection. In our study, diabetes was not a significant predictor of infections. Only atherosclerosis had a significant influence on the risk of infections.
Thank you for your time and valuable suggestions for our manuscript.
